# STOW: Discrete-Frame Segmentation and Tracking of Unseen Objects for Warehouse Picking Robots

**Yi Li**
University of Washington

**Muru Zhang**
University of Washington

**Markus Grotz**
University of Washington

**Kaichun Mo**
NVIDIA

**Dieter Fox**
University of Washington and NVIDIA

**Abstract:** Segmentation and tracking of unseen object instances in discrete frames pose a significant challenge in dynamic industrial robotic contexts, such as distribution warehouses. Here, robots must handle object rearrangement, including shifting, removal, and partial occlusion by new items, and track these items after substantial temporal gaps. The task is further complicated when robots encounter objects not learned in their training sets, which requires the ability to segment and track previously unseen items. Considering that continuous observation is often inaccessible in such settings, our task involves working with a discrete set of frames separated by indefinite periods during which substantial changes to the scene may occur. This task also translates to domestic robotic applications, such as rearrangement of objects on a table. To address these demanding challenges, we introduce new synthetic and real-world datasets that replicate these industrial and household scenarios. We also propose a novel paradigm for joint segmentation and tracking in discrete frames along with a transformer module that facilitates efficient inter-frame communication. The experiments we conduct show that our approach significantly outperforms recent methods. For additional results and videos, please visit website. Code and dataset will be released.

**Keywords:** Unseen Object Instance Segmentation, Unsupervised Multi Object Tracking, Zero-shot, Discrete Frames

## 1  Introduction

Object segmentation and tracking, a key perception task for robotic picking, is particularly important in warehouse environments, where millions of commodity items are organized daily on warehouse shelves for storage and categorization, as shown in Figure 1. Future intelligent robots must acquire strong perception capabilities to help human workers stow and fetch items from these shelves. These capabilities include detecting objects with diverse geometries in cluttered scenes and tracking them while other items are being added or picked up. Despite considerable research progress in this area, it remains notoriously difficult to detect and track unknown objects in highly cluttered environments.

Researchers commonly address this task using a *segment-then-track* paradigm [1, 2], which executes

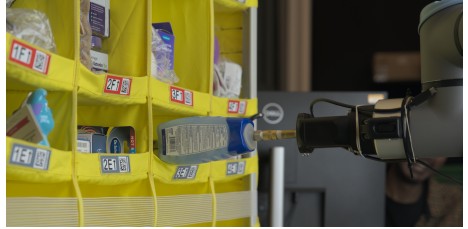

Figure 1: A densely packed shelf environment. The shelf holds objects from a wide array of categories. During the stowing process, a human operator may obscure the camera's view and rearrange the objects within the bin. The robot's task is to pick a specific object as directed by the given order index of the object's placement in the bin.

the two procedures sequentially. During segmentation, each frame is handled as an independent

7th Conference on Robot Learning (CoRL 2023), Atlanta, USA.

image on which advanced unseen object instance segmentation methods are applied [3, 4, 5, 6]. The subsequent tracking step leverages varied techniques [7, 8, 9, 10] to group masks of the same object across frames. However, this approach has inherent limitations. Segmentation methods struggle to resolve ambiguities because they cannot utilize information from other frames to enhance segmentation within each frame. Tracking lacks alternatives when segmentation fails, since its success heavily relies on consistent object appearance or location across consecutive frames. These drawbacks limit the paradigm's effectiveness when there are crowded scenes and discrete frames.

Our task resembles video instance segmentation (VIS), which involves segmenting and tracking objects in videos, with the leaderboard predominantly occupied by *simultaneous segmenting and tracking* methods [11, 12, 13]. This similarity suggests the potential to adopt their methods to realize an end-to-end solution in our task. However, their methods, mainly designed for videos with continuous frames, display subpar performance when faced with significant object movements between frames, a prominent challenge in our task.

We therefore introduce STOW, Discrete-Frame **S**egmentation and **T**racking of Unseen **O**bjects for **W**arehouse Picking Robots, a new framework for addressing challenges in our context. STOW consists of a new paradigm to jointly perform segmentation and tracking and a novel module, called the multi-frame attention layer, that facilitates efficient inter-frame communication. It succeeds in simultaneously achieving **high segmentation accuracy, high tracking accuracy, and high robustness to the sim-to-real gap.** Remarkably, even when trained exclusively on synthetic images, our method **significantly surpasses baseline on real data and live robot experiments.**

In summary, our main contributions are: (1) *Task formulation* for unseen object instance segmentation and tracking in discrete frames as well as realistic *synthetic data generation* and *real dataset data collection and manual labeling* for bins in the shelf and tabletop environments, facilitating research in this domain (2) *A new paradigm* to perform joint segmentation and tracking in discrete frames, along with a *new module, i.e., multi-frame attention*, that efficiently communicates information across frames (3) *Experiments conducted on real data and on a working robot* to verify our network's superior performance

## 2 Related Works

**Unseen object instance segmentation.** In computer vision, traditional object instance segmentation requires prior knowledge about the objects. In contrast, our work targets potentially unseen objects without such knowledge. Previous efforts, such as UCN [4], UOIS [3], and MF [5], addressed unseen object discovery in single-frame images using varied strategies, e.g., RGB-D feature embeddings and metric learning loss. A recent model, SAM [6], also demonstrates robust segmentation across a wide variety of objects by training on a large amount of data. While these works focus on segmenting and tracking in single-frame images, our research extends it to multiple frames.

**Video object segmentation.** Like the video object segmentation task [14, 15, 9], we track unseen objects in the test set. However, the VOS task assumes accessibility to an object's mask in one frame, which is not applicable in our task. For video object instance segmentation, previous works adhere to the *tracking by detection* paradigm, e.g., [16] and similarly [17, 18, 19], and often address the problem of multi-object tracking (MOT), i.e., the estimation of bounding boxes and identities of objects in consecutive RGB image streams. MOT tasks usually focus on tasks in traffic scenes and objects like people [20, 21] and vehicles.

**Video Instance Segmentation.** Our task diverges from existing VIS (Video Instance Segmentation) datasets [22, 23] in two main aspects. First, while VIS employs a closed-set category approach for detection/segmentation, our open-set problem adds complexity by recognizing instances regardless of class, as opposed to relying on learned patterns. Second, unlike VIS's focus on continuous, limited-changes video sequences, our dataset emphasizes tracking amid drastic changes between frames, making it ill-suited for benchmarking with VIS datasets.

**Unsupervised multi-object tracking.** The unsupervised video instance segmentation task introduced in DAVIS 2019 [24] bears similarity to Video Instance Segmentation (VIS), with a focus on

open-set category objects akin to our task. However, our approach diverges in two key aspects: (1) despite targeting unseen objects, DAVIS 2019 predominantly includes humans, vehicles, and animals, contrasting with the warehouse objects our study emphasizes, and (2) akin to VIS, it necessitates objects to exhibit continuous movement, thereby avoiding an ill-defined task.

**Image co-segmentation.** Our task is similar to object co-segmentation [25], which extracts recurring objects from an image pair or a set of images. While the goal of co-segmentation is to identify shared objects in a scene, we focus on instance segmentation and do not allow similar objects of the same class. Distinguishing between an object *instance* and an object *class* is a crucial requirement for industrial warehouses.

**Temporal Attention.** In temporal attention, our multi-frame method aligns with but uniquely stands out from prior works. Context R-CNN integrates per-RoI features from various frames to enhance current detections. Unlike its static approach and two-phase method, we embed temporal attention within each block, updating all frame object queries post each temporal attention. Meanwhile, [26] proposed module called Alignment-guided Attention (ATA) which apply temporal attention on patches with similar features through bipartite matching. Unlike ATA's 1D fixed-size patch focus, our technique employs all object queries across all images, capturing varied masks and accessing broader information.

## 3   Problem Formulation

In this work, we introduce the novel task of segmenting and tracking unseen objects given a series of input discrete image frames. Though challenging, this task has broad applications in the field of robotics and is particularly useful in warehouse environments, such as that shown in Figure 1. Scenes involving warehouse shelves can be exceedingly packed and cluttered, consisting of a vast assortment of items, including some that have not been previously encountered. Moreover, there may be temporal gaps between successive snapshots of the scene, during which human workers may place new objects, robots may retrieve existing items, and some objects may undergo changes in pose.

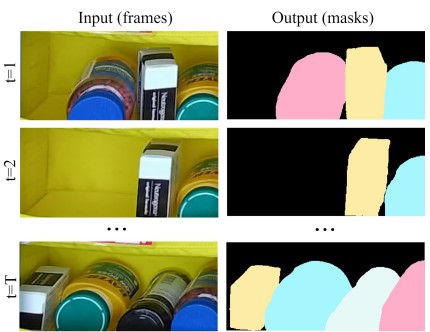

Figure 2: Task Illustration: The left column presents the images inputted into our network, while the right column showcases the expected segmentation and tracking outcomes. Identical colors indicate the same object.

Formally, we formulate the problem as follows. The input to the task is a sequence of images $\mathcal{I} = \{I_1, I_2, \cdots, I_T \mid I_t \in \mathbb{R}^{H \times W \times C_I}\}$, where $H$ and $W$ are the height and width of the images, respectively, and $C_I$ represents the number of channels, i.e., 3 for RGB images and 4 for RGB-D inputs. The task involves detecting, segmenting, and tracking $K_{\mathcal{I}}$ object instances that appear in these input images, where $K_{\mathcal{I}}$ may not be known beforehand. The output of the task is a set of binary object instance masks, $\mathcal{M}_t = \{M_t^1, M_t^2, \cdots, M_t^{K_{\mathcal{I}}} \mid M_t^i \in \{0, 1\}^{H \times W}, i = 1, 2, \cdots, K_{\mathcal{I}}\}$, corresponding to each input image $I_t \in \mathcal{I}$. Figure 2 shows the problem setting under consideration.

## 4   Method

Our system (Figure 3) uses query-based transformer architectures [27, 13] for object detection and segmentation, which we describe in Sec. 4.1. To enable the tracking of object instances across discrete image frames, we introduce the learning of additional object embeddings for tracking (Sec. 4.3) as well as a multi-frame attention layer to distinguish between identical or distinct object instances (Sec. 4.2). Sec. 4.4 discusses training specifics and loss functions.

### 4.1   Backbone Architecture

Following [27, 13], the backbone of our network consists of three components: (1) a ResNet-based image encoder, (2) a transformer-based object query decoder, and (3) several prediction heads tasked with determining object properties, such as the likelihood of object existence and object masks.

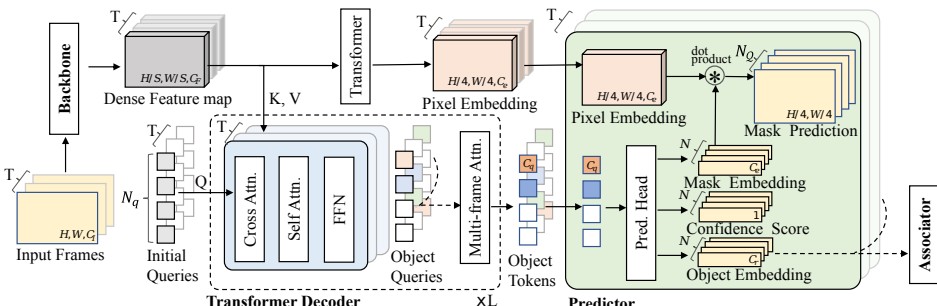

Figure 3: As outlined by a dashed rectangle, our transformer decoder ingests dense feature maps converted from input frames and produces object tokens for each image. These tokens predict confidence scores, mask embeddings for mask prediction, and object embeddings for association. We also introduce a novel "multi-frame attention" layer, which attends to object queries from all frames.

## 4.2 Multi-Frame Attention

**ResNet-based image encoder.** We use ResNet-50 [28] to transform every input frame $I_t \in \mathbb{R}^{H \times W \times C_I}$ ($t \in 1, 2, \ldots, T$) into a dense low-resolution feature map $F_t \in \mathbb{R}^{\frac{H}{S} \times \frac{W}{S} \times C_F}$. Here, $C_F$ denotes the channel dimension of the output dense feature map, while $S = 32$ is the down-sampling ratio used in this work.

**Transformer-based object query decoder.** We employ a DETR-like transformer decoder [29] that takes the produced dense feature map $F_t$ as input and learns to decode a set of $N_q$ object tokens $\{\mathbf{q}_t^1, \mathbf{q}_t^2, \cdots, \mathbf{q}_t^{N_q}\} \in \mathbb{R}^{C_q}$ as the outputs. Each object token contains the latent information necessary for tasks such as classification estimation, mask prediction, and tracking in discrete frames. Our transformer decoder consists of $L = 10$ transformer blocks; each block contains one cross-attention layer, one self-attention layer, one feed-forward layer, and one novel multi-frame attention layer that correlates object features across different image frames (Sec. 4.2).

**Prediction head for object masks.** To get a per-pixel segmentation mask $M_t^i \in \{0,1\}^{H \times W}$ for each object token $\mathbf{q}_t^i$, we first use a two-layer multilayer perceptron (MLP), which maps an input object token $q_t^i$ to a mask embedding $\mathbf{e}_t^i \in \mathbb{R}^{C_e}$; we then employ a multi-scale deformable attention transformer module (MSDeformAttn) [30] to convert the dense feature map $F_t$ to a pixel embedding map $P_t \in \mathbb{R}^{\frac{H}{4} \times \frac{W}{4} \times C_e}$. Here, $C_e$ denotes the channel dimensions used for the mask and pixel embeddings. We calculate the dot product between the object embedding $\mathbf{e}_t^i$ and the pixel embedding $P_t$ in order to obtain the mask prediction $\hat{M}_t^i$ at a reduced resolution of $\frac{H}{4} \times \frac{W}{4}$. Subsequently, a bilinear upsampling operator is applied to map $\hat{M}_t^i$ back to the original image resolution for the final mask prediction $M_t^i \in \{0,1\}^{H \times W}$.

**Prediction head for object existence scores.** Taking the object token $\mathbf{q}_t^i$ as input, we leverage a simple linear layer to estimate an object existence likelihood score $s_t^i \in [0, 1]$. In the context of unseen object instance segmentation, we are not concerned with precise target object categories, nor do we have access to this knowledge. This characteristic simplifies the task into a binary classification, the objective of which is to estimate the confidence score of whether each segment corresponds appropriately to an object.

## 4.3 Object Embedding for Tracking

In addition to the predicted object existence score and its segmentation mask in each frame, we add a new prediction head for object tracking embedding to enable the association of object tokens belonging to the same object from different input frames. Specifically, we employ a two-layer MLP to learn a mapping from the input object query $\mathbf{q}_t^i$ to another object embedding used for tracking $\mathbf{r}_t^i \in \mathbb{R}^{C_r}$. Here, $C_r$ represents the number of channels of the object embedding vector.

During the inference phase, we implement an associator to group object tokens with similar object embeddings. Specifically, we sequentially traverse each frame in the sequence. For each frame,

we retain only those object tokens $\mathbf{q}^i$ whose confidence scores surpass a predefined threshold $\delta_{\text{score}}$. For each input sequence, we maintain a trajectory bank, $\mathcal{T}$, that encompasses the trajectories of all observed objects. Each trajectory in the trajectory bank $\mathcal{T}^i \in \mathcal{T}$ consists of object tokens $\mathbf{q}^i$ that are considered to belong to the object $i$. Subsequently, we compute the similarity score between these selected object tokens and the previous trajectory using the following equation:

$$Sim(\mathbf{q}^i, \mathcal{T}^j) = \max(R(\mathbf{q}^i) \cdot R(\mathbf{q}_k^j)), \text{for } \mathbf{q}_k^j \in \mathcal{T}^j. \tag{1}$$

In this equation, $R$ is the function that transforms the object token $q^i$ into the object embedding $r^i$.

After this step, we use the Hungarian algorithm [31] to search for an optimal bipartite matching $\hat{\varrho}$ from all possible bipartite matchings $\mathcal{P}$ that can maximize the overall similarity between object tokens $\mathbf{q}$ and the trajectory bank $\mathcal{T}$:

$$\hat{\varrho} = \arg\max_{\varrho \in \mathcal{P}} \sum_i^{N_q} Sim(\mathbf{q}^i, \mathcal{T}^{\varrho(i)}). \tag{2}$$

We initialize the trajectory bank, $\mathcal{T}$, with an adequate number of false alarm tokens, $\mathcal{T}_{FA}$, each of which exhibits a constant similarity $\delta_{\text{match}}$ to all object tokens. The hyper-parameter $\delta_{\text{match}}$ can also be interpreted as a false alarm threshold in matching. Any predicted object tokens assigned to tokens from $\mathcal{T}_{FA}$ are assumed not to match any existing trajectory; thus, a new trajectory is opened for them.

Our tracking method also enables the possibility of handling multiple identical objects in the same scene. Specifically, object tokens corresponding to identical objects are likely to be recognized due to the expectation that object embeddings are near to them.

The previous modules we introduced work independently for each frame without any cross-frame information exchange. To facilitate efficient communication between frames, we introduce a new component, *the multi-frame attention layer*, into the transformer decoder.

The multi-frame attention layer is an extension of the self-attention layer, which operates on object queries from a single frame; it attends to object queries from all accessible frames. To illustrate, we denote the intermediate object queries after each feed-forward network as $\mathbf{X}_l^t$, where $l$ and $t$ indicate the index of transformer blocks and frames, respectively. A standard self-attention layer (with a residual path) computes the following (we omit the normalization term $\sqrt{d_k}$ here for simplicity):

$$\text{SelfAttn}(\mathbf{X}_l^t) = \text{softmax}(f_Q(\mathbf{X}_l^t) \cdot f_K(\mathbf{X}_l^t)^T)f_V(\mathbf{X}_l^t) + \mathbf{X}_l^t. \tag{3}$$

In contrast, our multi-frame attention layer computes:

$$\text{MultiFrameAttn}(\mathbf{X}_l^t) = \text{softmax}(f_Q(\mathbf{X}_l^t) \cdot f_K(\mathbf{X}_l)^T)f_V(\mathbf{X}_l) + \mathbf{X}_l^t. \tag{4}$$

Here, $\mathbf{X}_l$ represents the set of object queries from all frames $\mathbf{X}_l = \{\mathbf{X}_l^t, \text{ for } t = 1, 2, \ldots, T\}$. The functions $f_Q$, $f_K$, and $f_V$ are linear transformations that convert $\mathbf{X}_l$ or $\mathbf{X}_l^t$ into a $C$-dim space.

The multi-frame attention layer is positioned at the end of each transformer decoder block, which is repeated $L$ times in our network. Therefore, output object queries can incorporate the dense feature map of the current frame to update their prediction after communicating with object queries from other frames. Furthermore, the multi-frame attention layer is computationally efficient since it attends only to object queries from all frames, typically around 100 queries per frame vs the 1200 queries per frame used in Mask2Former-video [12] for images with size $640 \times 480$. Such efficiency lets it process long-term input sequences and large-scale images.

## 4.4   Training and Losses

Our model adopts the same loss function as utilized in Mask2Former [27] for classification and mask prediction. This includes the softmax cross-entropy loss, denoted as $\mathcal{L}_{\text{class}}$, for classification, along with binary cross-entropy loss, denoted as $\mathcal{L}_{\text{ce}}$, and the Dice loss, denoted as $\mathcal{L}_{\text{dice}}$, for mask prediction. To address tracking loss, we employ the contrastive loss $\mathcal{L}_{\text{contra}}$ used in DCN [32] along with the softmax loss (also referred to as the n-pair loss or InfoNCE loss) from CLIP [33]. For an object token $q_{\hat{t}}^{o_i}$, object tokens paired with the same object $o_i$ in different frames $q_t^{o_i}, t \neq \hat{t}$ are treated

as positive samples and pushed closer, while tokens assigned to different objects or backgrounds are treated as negative pairs and pushed away. See the supplementary material for more details.

Consistent with the approach in DETR [29] and Mask2Former [27], we leverage the Hungarian algorithm [31] to establish a bipartite matching between the predicted object tokens and ground truth that minimizes the overall loss. Notably, we exclude the tracking loss from the Hungarian algorithm's computation given that $\mathcal{L}_{softmax}$ is influenced by the object tokens contributing to this loss. The final loss is $\mathcal{L}_{\text{total}} = \lambda_{class}\mathcal{L}_{\text{class}} + \lambda_{ce}\mathcal{L}_{\text{ce}} + \lambda_{dice}\mathcal{L}_{\text{dice}} + \lambda_{\text{contra}}\mathcal{L}_{\text{contra}} + \lambda_{\text{softmax}}\mathcal{L}_{\text{softmax}}$.

# 5 Experiments

We assess our methodology in two typical environments—bin in a shelf and tabletop—both of which are representative settings in a multitude of warehouse and domestic applications. For each setting, we generate corresponding synthetic data for the training phase and collect and annotate real-world data for evaluation. We also integrate our approach into a bin-picking robotic system for practical experimentation.

We train our models separately on distinct synthetic datasets for the shelf and table environments. Each run exclusively uses one dataset and is evaluated against the corresponding real-world test set. Inference takes around 0.4 seconds for a 15-frame sequence on an RTX 2080Ti. Additional training details are in the supplementary material.

## 5.1 Dataset and Evaluation

**Synthetic data.** We construct mesh models for shelf and table bins using textures from the CC0 dataset [34], and object meshes from the Google Scanned dataset [35], consisting of over 1000 models. Excluding 70 with isolated parts, we utilize 900 objects for the training set and 100 for validation. Objects are randomly rotated and positioned to avoid collision, with no heavy occlusion in the shelf environment. In total, we generate around 10,000 sequences for the shelf, each with at least 2 packed frames, and 2,000 sequences for the tabletop, each containing 15 frames.

**Real-world data.** For real-world scenarios, we collect and manually label 44 sequences with 220 images for the shelf scenario and 20 sequences with 280 images for the tabletop scenario. Our dataset includes over 150 diverse objects. These range from relatively simple objects, such as boxes and bottles, to more complex ones, like transparent water bottles enclosed in plastic bags. In each sequence, we progressively add objects until either the bin is full or around 10 objects are placed on the table. Object rearrangement could occur between any two frames, leading to significant changes in object location and appearance.

**Evaluation metrics.** We adopt the evaluation method of the VIS challenge [22], a modified version of the MS-COCO metric [36]. In video instance segmentation, each object is represented by a series of masks, and the Intersection over Union (IoU) is calculated at the level of these mask sequences. To construct the Precision-Recall (PR) curve, the confidence threshold is systematically varied, with each threshold yielding a distinct data point on the curve. The area under the PR curve provides the Average Precision (AP).

In the context of this study, if not further specified, AP@0.5 denotes the average precision at an IoU threshold of 0.5. Similarly, AP@all represents the mean AP calculated over multiple IoU thresholds, specifically from 50% to 95% in 5% increments.

## 5.2 Results and Analysis

**Baseline methods.** We benchmark our approach against three state-of-the-art Video Instance Segmentation (VIS) methods: MinVIS [13], Mask2Former-video [12], and VITA [11]. To ensure an equitable comparison, all methods utilize ResNet-50 as the backbone, and the hyperparameters (such as batch size, maximum iterations, and the number of sampled frames) are standardized to match ours. Hence, all methods are trained on an identical number of images.

**Qualitative results.** As depicted in Table 1, our method notably outperforms existing VIS techniques, yielding an approximate 10% improvement in the shelf environment and a 20% increase in the table environment, even without using the multi-frame attention layer. Interestingly, MinVIS

| Method | Shelf | | Tabletop | |
|---|---|---|---|---|
| | AP@all | AP@0.5 | AP@all | AP@0.5 |
| MinVIS | 6.3 | 21.2 | 0.7 | 0.0 |
| Mask2Former Video | 35.0 | 66.1 | 27.7 | 56.7 |
| VITA | 42.7 | 70.1 | 26.6 | 55.0 |
| STOW (Ours) | **55.6** | **81.3** | **49.7** | **75.4** |

Table 1: Comparison between our method and leading video instance segmentation methods. Networks are trained on synthetic data and tested on real, unseen data. All use ResNet-50[28] and train on an identical number of images.

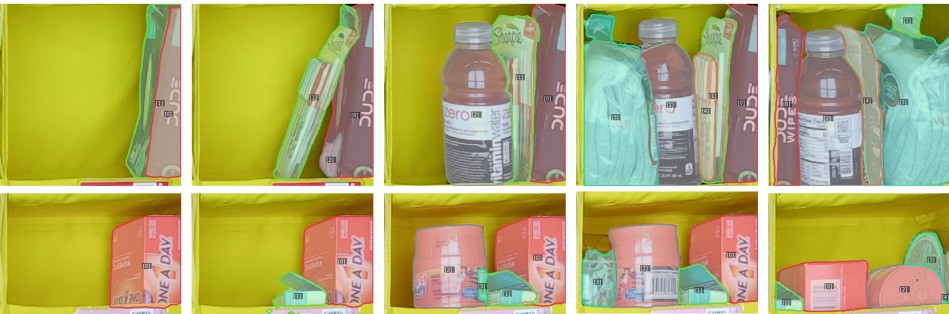

Figure 4: Visualized results for the shelf environment. Masks with the same color and index are associated and predicted as the same object by the network.

exhibits subpar performance in our task. We speculate that this is due to its reliance on the proximity between object tokens as a measure of similarity; this method presupposes that the changes between frames are minimal and that objects maintain similar locations and appearances. However, these conditions do not consistently hold in our task, potentially explaining the subpar performance.

**Quantative results.** We also show visualization results on a subset of real test data in Figure 4 and Figure 5. Our approach adeptly handles frame changes amid various types of noise. Despite significant movement and rotation between frames, the network successfully segments and tracks a broad array of object categories. It efficiently segments and continues tracking any new objects that are introduced. Figure 5 also demonstrates the method's robustness against backgrounds: it avoids predicting them as objects even though walls are not included in the training set. Supplementary materials contain additional results that contrast our approach with other methods.

### 5.3 Ablation Study

**Frame attention.** We evaluate the performance of our method with and without the cross-frame attention module on both the shelf and table scenes using ResNet-50 and Swin-T backbones. The cross-frame attention module yields consistent performance improvements across all configurations, as shown in Table 2.

**Sim2Real gap.** We evaluate our methods on the synthetic validation set, which contains objects not included in the training set, and compare it to the number we tested on the real test set. Results, shown in Table 3, reveal that other methods get results that are relatively to ours for the synthetic set, but their performance drops dramatically when we evaluate on the test set. This implies that they have difficulties solving the sim2real gap.

| multi frame | shelf | | tabletop | |
|---|---|---|---|---|
| | AP@all | AP@0.5 | AP@all | AP@0.5 |
| - | 51.8 | 78.7 | 44.4 | 68.5 |
| ✓ | **55.6** | **81.3** | **49.7** | **75.4** |

Table 2: With and without the multi-frame attention layer. The left column denotes whether we incorporate multi-frame attention in this experiment. All other hyper-parameters remain the same. Use of the frame attention layer boosts both shelf and tabletop environment performance by ~5%.

### 5.4 Real Robot Applications

We integrated our visual perception technique into an autonomous shelf-picking system [37]. The system's multi-component software architecture is managed by a state machine. The system setup uses a UR16e industrial robot situated in front of an industrial warehouse shelf filled with objects. Within the robotics community, it's standard to

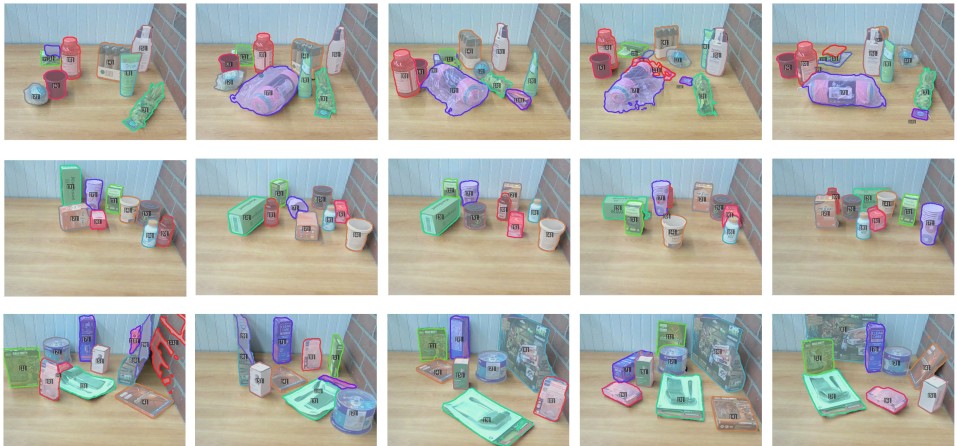

Figure 5: Visualized results for the table environment. Masks with the same color and index are associated and predicted as the same object by the network.

address perception problems by combining unseen instance segmentation methods with other established techniques, as seen in [1],[38], [39], [40]. Other than VITA [11] in Table 1, we also combine UCN [4], a staple in unseen instance segmentation, with SIFT [7], a renowned keypoint extraction method, reflecting the conventional solution for this challenge. The center of the mask is used as the grasping point for suction cap. Our evaluation protocol employs a fixed set of diverse items, stowed at specific locations and orientations within the bins, to ensure reproducibility and comparability of results since performance can fluctuate with different item configurations and inherent system stochasticity.

We testing each method with 50 trials across different levels of difficulty, involving over 100 objects. For UCN [4]+SIFT [7], the perception success rates stand at 56%. With our STOW method, this rate increase to 76%.

| method | synthetic | | real | |
|---|---|---|---|---|
| | AP@all | AP@0.5 | AP@all | AP@0.5 |
| MinVIS[13] | 0.3 | 2.6 | 0.7 | 0.0 |
| M2F-V[12] | 71.6 | 83.7 | 27.7 | 56.7 |
| VITA[11] | 69.4 | 81.9 | 26.6 | 55.0 |
| Ours | **74.1** | **89.3** | **49.7** | **75.4** |

## 6 Limitations

Our method, while effective, has limitations in handling highly cluttered environments and complex objects. False positives and negatives occur in object detection, especially in intricate settings. The segmentation process can result in over- or under-segmentation due to complex object boundaries and textures. In object track-

Table 3: Ablation study on solving the sim2real gap. After training on synthetic tabletop training set, we separately evaluate each method on a synthetic tabletop validation set and a real tabletop test set; note that objects in the synthetic validation set are not included in the synthetic training set.

ing, we sporadically encounter mistracking incidents and occasional failures to distinguish between multiple objects. Refer to the supplementary material for detailed analyses and examples of these limitations.

## 7 Conclusion

In this paper, we introduce the task of segmenting and tracking unseen objects in discrete frames which is widely used in robotics tasks but under investigation. We formulated the problem and collected both synthetic and real datasets. We also propose a novel paradigm for joint segmentation and tracking, incorporating multi-frame attention for better inter-frame communication. Even when trained solely on synthetic data, our method adeptly handles clustering and large movements in real-world sequences. Our innovative approach excels in segmenting and tracking within both shelf and tabletop settings, surpassing state-of-the-art techniques with a 10%-20% improvement in AP in real-world scenarios and more than 20% success rate in robot experiments.

**Acknowledgments**

This research is funded by the UW + Amazon Science Hub as part of the project titled, "Robotic Manipulation in Densely Packed Containers." We would like to thank Dr. Michael Wolf from Amazon for valuable discussions. We further would like to thank our students Sanjar Normuradov and Soofiyan Atar

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

# A    Dataset Detail

## A.1    Synthetic Data

We built a synthetic dataset using high-quality household models from the GoogleScanned dataset[35] with two typical settings: a) Shelf and b) Tabletop.

**Shelf environment.**    In shelf environments or other bin-based object arrangements, the objects are akin to books and are constrained to a shortest-dimension-faces-outward orientation. This scheme ensures that each object is guaranteed to have at least one visible face, but it also leads to significant occlusion among objects. The camera is positioned at the front of the bin to capture images of the scene, subject to random perturbations in the location that inject noise into the data.

Given that each bin contains a maximum of 3 to 5 objects, segmentation and tracking tasks become trivial if the scene contains fewer than 3 objects. To address this issue, image frames are generated only when the bin is nearly full. We leverage approximately 900 objects sourced from the Google Scanned dataset, resulting in a training set of approximately 9000 image pairs. We use the remaining

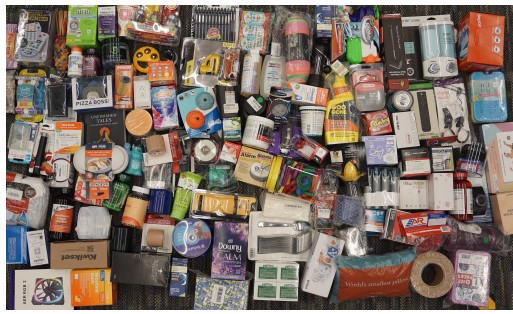

Figure 6: Some objects used during the evaluation. Objects vary greatly in shape and physical properties, with some being partially transparent or wrapped in a bag.

100 objects to generate approximately 1000 image pairs for the test set. Each image pair may exhibit the introduction of a new object in addition to existing objects undergoing a flipping operation or relocation with a certain probability.

**Tabletop environments.** Generating datasets of objects placed on a table requires different settings given the absence of walls and typically larger surface area than in a bin-based object arrangements. As a result, we adopt an alternative strategy for dataset generation. Specifically, each sequence consists of 15 images, with the first 10 images incrementally introducing new objects while shuffling existing objects between frames. No new objects are added in the final 5 frames, though the shuffling of existing objects persists. Due to the random placement of objects on the table, instances of full occlusion may occur in certain frames and subsequently reappear in subsequent frames.

To construct our training and testing datasets, we utilize 900 objects sourced from the Google Scanned dataset, producing 2000 sequences for the training set, with the remaining 100 objects used to generate 500 sequences for the test set.

## A.2  Real Data

As we did for the synthetic evaluation, we split the evaluation into shelf and tabletop environments, the most common real-world scenarios encountered. To evaluate our method on challenging real-world scenarios, we need a large variety of objects; Figure 6 depicts some of the objects used during the tablefop evaluation. For shelf environments, we use an Azure Kinect RGB-D sensor, and for tabletop ones we use an Intel Realsense D455 camera. Camera distance ranges from 1 to 1.5 meters. Each time an object is placed on the table or in a new bin, a new image is captured. Objects can be rearranged to maximize space utilization as they are placed in the scene. After all the objects are placed in the scene, we also displace the objects for a more refined evaluation. Camera images are manually labeled using the interactive segmentation of the object tracking framework XMem [9]. We collected and annotated more than 280 images with more than 150 different objects for the tabletop scenario and 220 images for the shelf scenario.

## B  Training and Inference Details

### B.1  Training Details

We set the maximum number of iterations to 16k using an initial learning rate of 1e-5, which was then dropped by 0.1 after 14k iterations. The number of classes is set to 1 since we are aiming to handle unseen objects. For the shelf dataset, we trained our network with a batch size of 32 and leveraged 2 frames from each sequence; for the table dataset, we set the batch size to 8 and randomly selected 4 frames from each sequence. To enhance the diversity of our dataset, we applied random color jittering and rotation to the input before feeding it to the network. The training process was executed on a single NVIDIA A-40 GPU and took approximately 13 hours.

During the training phase, we excluded the initial predicted object embedding, which was directly generated from the query feature. Additionally, when handling negative queries, we adopted a more selective approach by considering only queries whose IoU with any ground truth was lower than 0.6 rather than regarding all unmatched queries as negatives. This was motivated by the lack of clarity regarding which patches truly represent objects in unseen object settings (in contrast to close-set settings).

## B.2 Associator

We show below an example of code demonstrating how to associate object tokens from a new frame with the trajectory bank built in previous frames. In implementation, we set $\sigma_{score} = 0.6$ and $\sigma_{match} = 0.2$ (similarity ranged in $[-1, 1]$).

```python
def associate_one_frame(traj_bank, object_tokens_cur_frame, delta_score,
    delta_track):
    object_tokens = [x for x in queries_this_frame if x['score']>
        delta_score]
    num_trackers = len(traj_bank)
    Nq = len(object_tokens)
    similarity = torch.ones(num_trackers+Nq, num_pred)*delta_track

    # Extract object embedding from current frame's object tokens
    obj_embed = torch.stack([x['obj_embed'] for x in object_tokens])

    # Compute similarity between object embedding of trajectory and
        current frame's object tokens
    for traj_idx, traj in enumerate(traj_bank):
        traj_obj_embed = torch.stack([x['obj_embed'] for x in traj])
        sim = traj_obj_embed @ obj_embed
        similarity[traj_idx] = sim.max(dim=0)[0]

    # Perform Hungarian matching to find bipartite matching which have
        hightest similarity
    traj_indices, obj_token_indices = hungarian_matching(-similarity)

    # Update tracker
    for traj_idx, token_idx in zip(traj_indices, obj_token_indices):
        if traj_idx > num_trackers:
            # if it is not matched with any existing trajectory
            traj_bank.append([object_tokens[token_idx]])
        else:
            traj_bank[traj_idx].append(object_tokens[token_idx])
    return traj_bank
```

## B.3 Loss

We keep the loss function that Mask2Former used for classification and mask prediction, which means binary cross entropy and dice loss for mask prediction and softmax cross entropy loss for classification.

For the object embedding head, we also use two losses: contrastive loss and softmax loss (or n-pair loss and InfoNCE loss).

**Contrastive Loss.** We use contrastive loss modified from DCN[41] with hard-negative scaling from [32].

$$\mathcal{L}_{\text{matches}}(Q) = \frac{1}{N_{\text{matches}}} \sum_{N_{\text{matches}}} D(q_{t1}^{o_i}, q_{t2}^{o_i})^2 \tag{5}$$

$$\mathcal{L}_{\text{non-matches}}(Q) = \frac{1}{N_{\text{hard-neg}}} \sum_{N_{\text{non-matches}}} (0, M - D(q_{t1}^{o_i}, q_{t2}^{o_j})_{i \neq j}) \tag{6}$$

$$\mathcal{L}(Q) = \mathcal{L}_{\text{matches}}(Q) + \mathcal{L}_{\text{non-matches}}(Q), \tag{7}$$

where

$$N_{\text{hard-negatives}} = \sum_{N_{\text{non-matches}}} \mathbb{1}(M - D(q_{t1}^{o_i}, q_{t2}^{o_i}) > 0). \tag{8}$$

Here, $Q$ denotes all object tokens from images, and $q_t^{o_i}$ denotes the object tokens assigned to object $o_i$ in frame $t$. $M$ is the margin parameter used to ensure that non-matched pairs have a distance of at least $M$ apart. The distance function $D$ is the cosine distance function, as in UCN [4], which is defined as:

$$D(q^i, q^j) = \frac{1}{2}(1 - r^i \cdot r^j). \tag{9}$$

Here, $r^i = \frac{f(q^i)}{|f(q^i)|}$ is the object embedding of object token $i$, which is computed by first forwarding the query to a linear layer $f$ and then normalizing it to a unit vector. To expedite the training process, we selectively incorporate a subset of negative queries to contribute to the contrastive loss, thereby enhancing its efficiency.

An illustration of the contrastive loss is shown in Figure 7. Assuming that three frames are sampled from a sequence during training, the contrastive loss will be computed between all frames. In the figure, matched pairs are denoted by dark gray and apply loss according to Equation 5, while non-matched pairs are denoted by light gray and apply loss according to Equation 6.

**Softmax Loss** We also modified the N-pair/InfoNCE loss used in CLIP[33].

$$\mathcal{L}_{\text{softmax}}(t) = -\sum_{k \in Q^+} \sum_{i \in O(t)} \frac{\exp(r_t^k \cdot r_t^i \cdot e^\tau)}{\sum_{j \in Q_t} \exp(r_t^k \cdot r_t^i \cdot e^\tau)} \tag{10}$$

$$\mathcal{L}_{\text{softmax}} = \frac{1}{T} \sum_{t \in 1, \cdots, T} \mathcal{L}_{\text{softmax}}(t), \tag{11}$$

where $Q^+$ denotes all positive queries, $O(t)$ denotes all objects in frame $t$, and $Q_t$ denotes all queries in frame $t$. This is also shown in Fig. 7-b, where the label of each row corresponds to the index of the query assigned to the same object. If there are $n$ identical objects in the same frame, the softmax loss should be extended by copying all queries in this frame $n$ times, each time keeping only one query for that object. This allows queries containing the same object to be converted into multiple query sets, each consisting of only the target object.

Thus, the final tracking loss can be represented as

$$\mathcal{L}_{\text{track}} = \lambda_{\text{contra}} \mathcal{L}_{\text{contra}} + \lambda_{\text{softmax}} \mathcal{L}_{\text{softmax}}. \tag{12}$$

## C  Detailed Analysis

### C.1  Sim-to-real Gap

Results are shown in Table 4. The experiment is conducted in the tabletop environment with the same setting as in main manuscript. We see the following from

(1) The Image AP results from both the synthetic validation set and real test set demonstrate that MinVIS outperforms Mask2Former Video and VITA. This indicates that the approach of

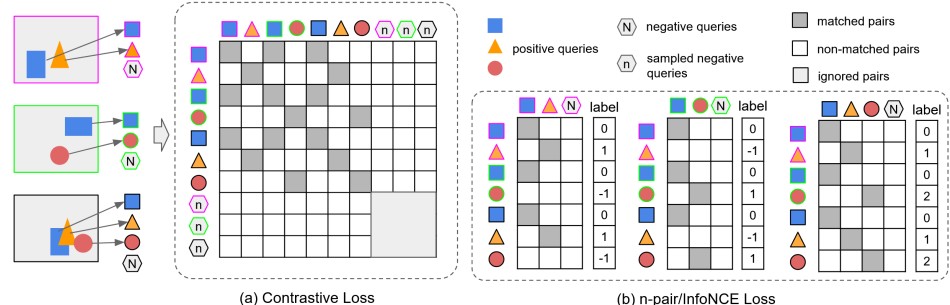

(a) Contrastive Loss          (b) n-pair/InfoNCE Loss

Figure 7: Tracking loss. In this example, three frames are sampled from a sequence, denoted with different border colors. Object tokens that match the objects in the images are represented by a blue square, an orange triangle, and a red circle. The hexagon denotes background object tokens that do not match to any objects. (a) the contrastive loss is computed between all frames, where matched pairs (dark gray) apply loss using Equation 5, non-matched pairs (white) apply loss using Equation 6, and ignored pairs (light gray) do not contribute to the loss. (b) the n-pair/InfoNCE loss is computed over all positive queries and queries from each frame. Equivalent to using a softmax cross-entropy while setting the label of the index of queries assigned to the same object.

| method | syn., video AP | | syn., image AP | | real, video AP | | real, image AP | |
|---|---|---|---|---|---|---|---|---|
| | AP@all | AP@0.5 | AP@all | AP@0.5 | AP@all | AP@0.5 | AP@all | AP@0.5 |
| UCN [4] | - | - | 67.3 | 82.9 | - | - | 52.4 | 86.6 |
| MinVIS [13] | 0.3 | 2.6 | 82.4 | 91.8 | 0.7 | 0.0 | 54.5 | 72.7 |
| Mask2Former Video [12] | 71.6 | 83.7 | 72.8 | 82.6 | 27.7 | 56.7 | 38.9 | 57.3 |
| VITA [11] | 69.4 | 81.9 | 70.6 | 80.2 | 26.6 | 55.0 | 41.4 | 63.0 |
| Ours | **74.1** | **89.3** | **87.6** | **95.3** | **49.7** | **75.4** | **80.1** | **97.6** |

Table 4: Evaluation of SOTA VIS methods on the unseen object instance segmentation task. "syn" indicates evaluation on a synthetic tabletop dataset; "real" denotes evaluation on a real-world table-top dataset. The evaluation metrics include "video" and "image" for assessing performance in different contexts (as described in subsection B.1). We see that MinVIS exhibits superior performance in detection, while Mask2Former Video and VITA excel in matching. Remarkably, our proposed method harnesses the strengths of both approaches, surpassing all evaluated methods in overall performance.

Mask2Former Video and VITA, which utilizes a single object token to predict object masks across an entire sequence, is less effective than using distinct object tokens for each frame. Consequently, it is difficult for object tokens to efficiently manage discrete frames with considerable movement and appearance variations.

(2) A significant decline is apparent in the Image AP and Video AP results on the real test set for MinVIS. This suggests a suboptimal tracking performance when applying object tokens directly.

(3) Notably, our method experiences a less dramatic drop in performance, as evidenced by the Video AP results on both the synthetic validation set and the real test set. This suggests that our method is more adept at managing the simulation-to-reality gap.

# D    Failure Cases

The efficacy of our method is sometimes compromised in scenarios characterized by crowded scenes or significant alterations in object appearance, as illustrated in Figure 10. We categorize these shortcomings into two principal groups: segmentation failures and tracking failures.

Segmentation failures arise from issues such as:

- *Under-segmentation*: This occurs when objects have similar colors or lack clear borders, leading to a blending of distinct entities.

- *Over-segmentation*: In this case, a single object is erroneously identified as multiple entities due to recognition failures.

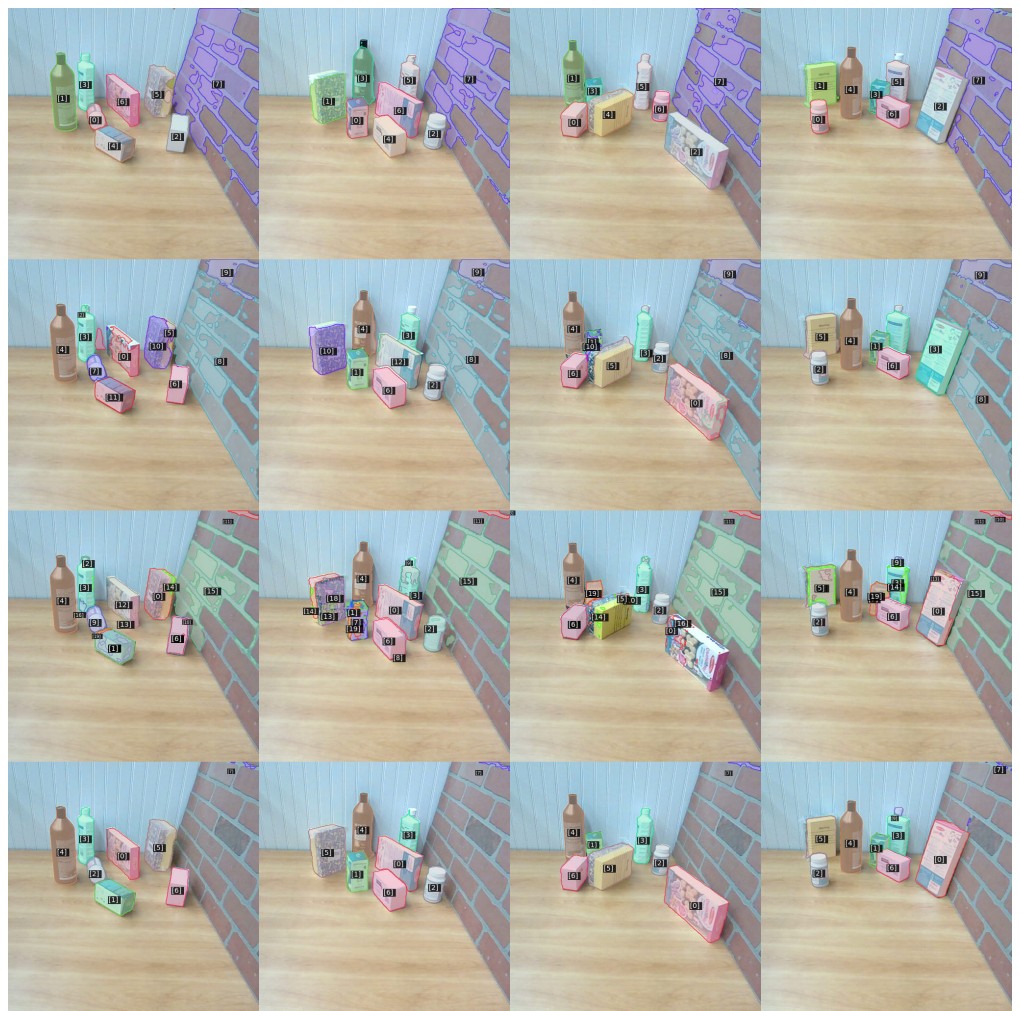

Figure 8: Results from different methods on the tabletop dataset. Methods ordered from top to bottom: MinVIS, Mask2Former-Video, VITA, and Ours (STOW).

- *Detection failure*: Here, an object is entirely missed, leading to its absence in the segmented output.

Tracking failures, on the other hand, include:

- *Mismatch*: This involves incorrect associations between objects across frames or confusion arising from similar-looking distractors.
- *Mistrack*: In these instances, the algorithm fails to consistently identify the same object, resulting in tracking inconsistencies.

In both categories of failure, the complexity of scene compositions and variations in object appearances are pivotal factors that undermine the performance of our tracking method. We are devoted to exploring advanced strategies to mitigate these limitations, aiming for enhanced robustness in diverse and dynamic environments.

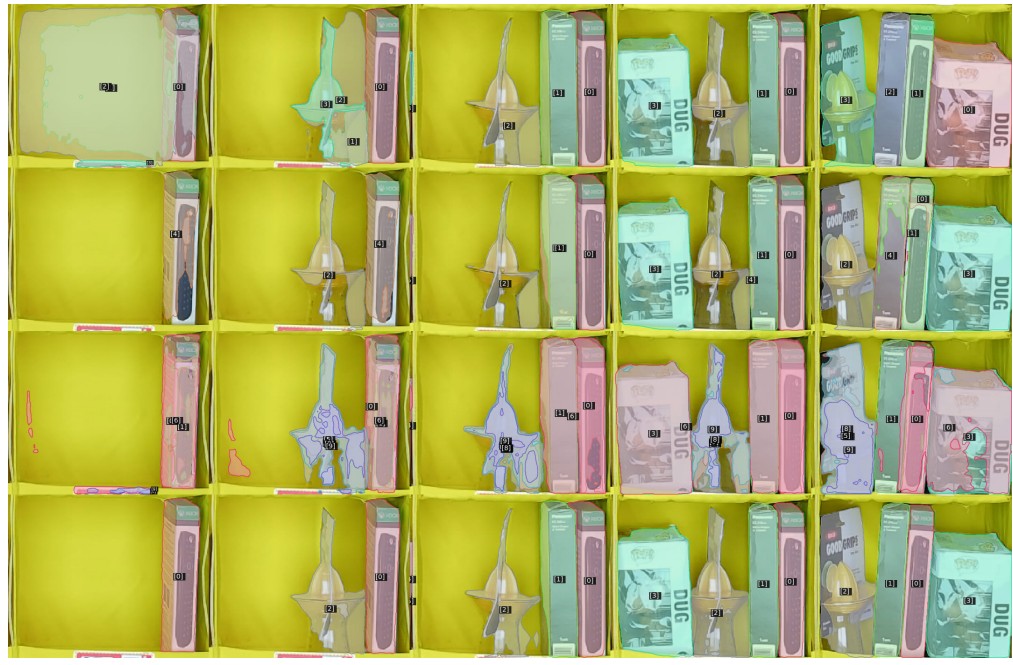

Figure 9: Results from different methods on the bin dataset. Methods ordered from top to bottom: MinVIS, Mask2Former-Video, VITA, and Ours (STOW).

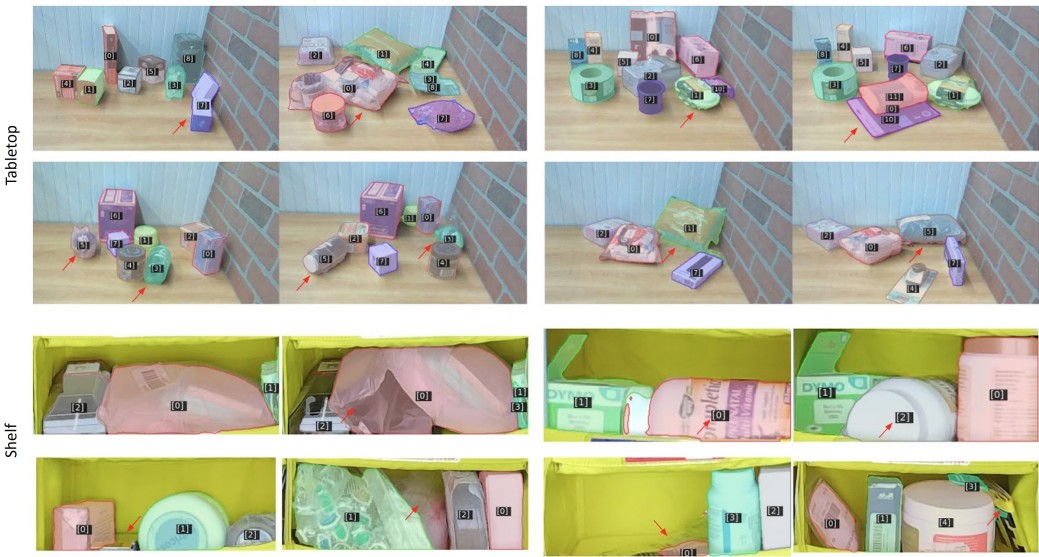

Figure 10: Failure Cases Illustrated. In the Tabletop settings (top 2 rows), we observe under-segmentation (top left), over-segmentation (top right), mismatch (bottom left), and mistrack (bottom right). In the Shelf settings (bottom two rows), the failures include a combination of under-segmentation and failure to detect new objects (top left), failure to track and mismatch (top right), failure to detect (bottom left), and failure to segment (bottom right).

