# OpenReview forum: "STOW: Discrete-Frame Segmentation and Tracking of Unseen Objects for Warehouse Picking Robots"
_robot-learning.org/CoRL/2023/Conference — CoRL 2023 Poster_

### Official Review · Reviewer_izjm · 2023-07-20

**Confidence:** 3
**Originality:** Good
**Technical Quality:** Good
**Clarity Of Presentation:** Very Good
**Impact:** 3

**Recommendation:**

Weak Accept: I recommend accepting the paper, but will not argue for my recommendation if the majority of other reviewers have a different opinion.

**Review:**

Strengths
* Overall the presentation and contributions are clear.
* The architecture introduced in the paper is interesting and makes sense for this segmentation and tracking task.
* Ablation studies show that the multi-frame attention helps for shelf and table scenes.
* The synthetic + real dataset introduced seem useful for these sorts of industrial warehouse tasks.
* Real robot results including videos.
* Failure cases described in appendix.

Weaknesses
* It seems that results are only reported on the datasets constructed for this task. It would also be interesting to compare to other methods using other datasets, even not specific to robotics.
* While some results are mentioned in the real world robot setting, there seems to have been only two test cases for evaluating the method for real robot tasks.


**Quality Of The Limitations Section:**

Limitations are addressed clearly

**Questions For Rebuttal:**

* A more thorough study on failure cases may be interesting, to better understand where methods like this may fail. E.g. Are there scenes that are too crowded for proper segmentation/tracking?
* Will the code and dataset be released?


**Robotics Focus:**

Sufficient demonstration on hardware

**Summary Of Paper:**

The paper introduces a transformer based architecture for segmentation and tracking that enables tracking across discrete image frames. This allows the system to work in packed and cluttered scenes such as warehouse shelves, and also handles unseen objects between frames. Compared to other state of the art video segmentation models, this methods improves performance for shelf and tabletop synthetic datasets. The paper also introduces a synthetic and real dataset.

**Summary Of Recommendation:**

I am recommending a weak accept. The overall system and architecture seem sound - while this work does not seem to contain any major advances, there is some novelty in the architecture that shows better results compared to other methods.

---

### Official Review · Reviewer_Bc1q · 2023-07-20

**Confidence:** 4
**Originality:** Good
**Technical Quality:** Very Good
**Clarity Of Presentation:** Excellent
**Impact:** 3

**Recommendation:**

Weak Accept: I recommend accepting the paper, but will not argue for my recommendation if the majority of other reviewers have a different opinion.

**Review:**

The paper overall is quite strong and presents a novel approach to tracking objects across frames with significant differences. I believe the paper has the following strengths.

- Very well written and clear figures.
- Solid proposed architectural solution to the problem.
- A good set of experiments and ablations that demonstrate the performance of the method.
- Real robot experiments using the method.

I have the following suggestions/comments for the authors as well.

- I was wondering whether the multi-frame attention is truly a novel contribution of this work. What is unique about the multi-frame attention layer compared to temporal attention from past papers, with just a few examples below.
    - https://arxiv.org/abs/1912.03538
    - https://arxiv.org/abs/2210.00132
- I’m curious as to how this would compare to a straightforward/naive approach of using a frozen segmentation network (e.g. SAM) combined with some pre-trained image embedding on the masked objects from earlier frames and some bipartite matching with dot products.
- The examples shown in the video all seem quite straightforward. I’m wondering if there are simpler methods (like the point above) that could work equally well for this. Is there really a need for an end-to-end architecture for this problem? If so, can you demonstrate success on examples where a simpler method wouldn’t work?

**Quality Of The Limitations Section:**

Additional details required

**Questions For Rebuttal:**

- Is it possible to move the architecture figure a bit earlier in the paper? It is useful to refer to it as the architecture is being described.
- How is the multi-frame attention different from past papers using temporal attention?
- Why is a different baseline used for real-world physical experiments compared to the perception-only experiments?
- How does this do when objects are lying down vs standing upright? Is the method able to consider this dramatic variation in pose as belonging to the same object?
- Is there any explanation for why your method does better transferring to real compared to other methods?
- In related work, co-segmentation was mentioned but the difference between that and this method is not too clear.
- What is the performance difference between a simpler method mentioned in the review above (not end-to-end) versus this end-to-end approach.

**Robotics Focus:**

Sufficient demonstration on hardware

**Summary Of Paper:**

This method presents an architecture for object segmentation and tracking across frames. While prior methods include video-based object tracking assuming minimal movement between frames, this method makes no such assumptions. It leverages attention across time along with bipartite matching of embeddings into the past, in order to track objects, including those that disappear and reappear, over time.

**Summary Of Recommendation:**

Overall, the paper is solid. However, I have some concerns about the claimed novelty and performance compared to some simpler pipelines, which I hope the authors will address in the rebuttal.

---

### Official Review · Reviewer_ySeC · 2023-07-22

**Confidence:** 2
**Originality:** Fair
**Technical Quality:** Very Good
**Clarity Of Presentation:** Very Good
**Impact:** 2

**Recommendation:**

Weak Reject: I recommend rejecting the paper, but will not argue for my recommendation if the majority of other reviewers have a different opinion.

**Review:**

The paper is well-written and easy to understand. The authors do a good job of providing background information on different datasets and problems related to instance segmentation. The method is well-described and includes some novel components, such as the multi-frame attention.

However, I'm having trouble identifying a significant difference between this problem setting and video instance segmentation (VIS), which serves as a popular benchmark for segmenting videos and comes with a few public datasets (YouTube-VIS 19, YouTube-VIS 21, OVIS). The introduction says "... these methods are mainly designed for videos with continuous frames. As a result, they display subpar performance when faced with significant object movements between frames, a prominent challenge in our task." Based on this, my understanding is that the dataset used in this paper is a low frame rate version of the VIS task. This observation seems to be supported by the fact that all the baseline comparisons are made against VIS models.

If the only difference between this dataset and VIS is the frame rate, then I believe this model should be compared to downsampled VIS datasets, which are comparatively much larger and more diverse. By evaluating the proposed model on larger datasets, it would better support the claim of its superiority at lower frame rates. If the proposed model architecture does perform better under lower frame rates, it would be an important finding that could be strengthened with evaluations on larger datasets.

**Quality Of The Limitations Section:**

Limitations are addressed clearly

**Questions For Rebuttal:**

What are the differences between this dataset and VIS?
Why wasn't this model architecture benchmarked on existing VIS datasets?

**Robotics Focus:**

Sufficient demonstration on hardware

**Summary Of Paper:**

This paper presents STOW, a dataset and model architecture for tracking unseen objects over time. The dataset consists of scenes from industrial and household scenarios. The model architecture is transformer based image model. Experimental results show improvement over baselines.

**Summary Of Recommendation:**

I think the paper is well written and has some novel model architecture contributions. However, I'm unable to see a difference between STOW and a low-frame rate version of VIS. If there are more differences between the datasets than frame rate, the authors should better explain the difference. At the very least, I think VIS deserves a paragraph in related work. Otherwise, the should include downsampled versions of VIS datasets in their model comparisons.

---

### Decision · Program_Chairs · 2023-08-30

**Decision:**

Accept (Poster)

**Comment:**

**Strengths**:

- The paper is well-written and easy to understand.

- The architecture introduced in the paper is interesting and novel, e.g. the multi-frame attention, and makes sense for this segmentation and tracking task.

- A good set of experiments and ablations that demonstrate the performance of the method.

- The synthetic + real dataset introduced seem useful for these sorts of industrial warehouse tasks.

- Real robot experiments using the method.

**Weaknesses**:

- Results are only reported on the datasets constructed for this task. It would also be interesting to compare to other methods using other datasets, even not specific to robotics.

- Lack of a section on limitation discussion.


Post-rebuttal: The authors have made great responses to the concerns from the reviewers. Additional results also clarify the difference between STOW and a low-frame rate version of VIS. The authors are encouraged to publish the code and dataset with the final version. Also please revise the paper in line with the rebuttal discussion for the camera-ready submission.